# Effect of Bainite to Ferrite Yield Strength Ratio on the Deformability of Mesostructures for Ferrite/Bainite Dual-Phase Steels

**DOI:** 10.3390/ma14185352

**Published:** 2021-09-16

**Authors:** Gui-Ying Qiao, Zhong-Tao Zhao, Xian-Bo Shi, Yi-Yin Shan, Yu Gu, Fu-Ren Xiao

**Affiliations:** 1Key Lab of Applied Chemistry of Hebei Province, School of Environment and Chemical Engineering, Yanshan University, Qinhuangdao 066004, China; qiaoguiying@ysu.edu.cn; 2Key Lab of Metastable Materials Science&Technology, Hebei Key Lab for Optimizing Metal Product Technology and Performance, College of Materials Science Engineering, Yanshan University, Qinhuangdao 066004, China; zhaozhongtao@hotmail.com; 3Institute of Metal Research, Chinese Academy of Sciences, Shenyang 110072, China; xbshi@imr.ac.cn; 4School of Mechanical and Electrical Engineering, Zhoukou Normal University, Zhoukou 466000, China; 202guyu@163.com

**Keywords:** F/B dual-phase steel, strength difference in bainite to ferrite, deformability, deformation compatibility, simulation

## Abstract

The strength and plasticity balance of F/B dual-phase X80 pipeline steels strongly depends on deformation compatibility between the soft phase of ferrite and the hard phase of bainite; thus, the tensile strength of ferrite and bainite, as non-negligible factors affecting the deformation compatibility, should be considered first. In this purely theoretical paper, an abstract representative volume elements (RVE) model was developed, based on the mesostructure of an F/B dual-phase X80 pipeline steel. The effect of the yield strength difference between bainite and ferrite on tensile properties and the strain hardening behaviors of the mesostructure was studied. The results show that deformation first occurs in ferrite, and strain and stress localize in ferrite prior to bainite. In the modified Crussard-Jaoul (C-J) analysis, as the yield strength ratio of bainite to ferrite ($\sigma_{y,B}/\sigma_{y,F}$) increases, the transition strain associated with the deformation transformation from ferrite soft phase deformation to uniform deformation of ferrite and bainite increases. Meanwhile, as the uncoordinated deformation of ferrite and bainite is enhanced, the strain localization factor (SLF) increases, especially the local strain concentration. Consequently, the yield, tensile strength, and yield ratio (yield strength/tensile strength) increase with the increase in $\sigma_{y,B}/\sigma_{y,F}$. Inversely, the strain hardening exponent and uniform elongation decrease.

## 1. Introduction

With the rapid increase in demand for clean energy, such as natural gas, long-distance gas transportation pipelines are developing towards larger diameters and higher operating pressures to maximize transport efficiency, promoting the rapid development of high-strength pipeline steel. Thus, high-strength X80 grade steels have been applied in high-throughput long-distance gas transportation pipelines [1,2,3]. Although the traditional acicular ferrite (or low-carbon bainite) high-strength pipeline steels have good strength, toughness, and weldability balance, their plasticity does not meet the requirements of long-distance gas transportation pipelines to prevent deformation collapse caused by geological and ocean current movement in hazardous geological areas. Therefore, plasticity improvement of pipeline steel has become the focus of research on high-strength pipeline steel [4,5].

The mechanical properties of steel strongly depend on its microstructure. The duplex-phase microstructure steels take advantage of the mechanical properties of different steels′ mechanical properties to obtain better comprehensive mechanical properties than single-phase structured steel. Thus, the duplex-phase design has become the first option in improving the comprehensive mechanical properties of steels, and a new type of ferrite/bainite (F/B) dual-phase X80 pipeline steel with excellent deformability has been proposed and widely researched [6,7,8].

Usually, the F/B dual-phaseX80 pipeline steels are produced by an advanced thermo-mechanical controlled process (TMCP) [9,10]. The TMCP parameters, including reheating temperature, beginning rolling temperature, rolling reduction, finishing rolling temperature, beginning quick cooling temperature, cooling rate, and finishing cooling temperature, strongly affect the microstructure of F/B dual-phase X80 pipeline steels, such as the volume fraction, distribution, morphology and mechanical properties of each phase [9,10,11,12,13]. Thus, the F/B steels’ microstructure effects on mechanical properties, strain hardening behavior, microscopic deformation, and failure mechanism have been studied [14,15]. However, those research works focused on the volume fraction of bainite effects on the strength, yield ratio, uniform elongation, and strain hardening exponent. On the contrary, our research has shown that the mechanical properties andstrainhardening behavior of the F/B dual-phase X80 pipeline steel are independent of the volume fraction of bainite [16] and dependent on the morphology and distribution of bainite [17]. This is because the deformation ofthe low-strength ferrite occurs more easily during plastic deformation; concomitantly, strain and stress concentration occurs at the F/B interface, and the plastic deformation occurs in bainite to accommodate the high deformation properties of ferrite when the stress concentration near the F/B interface is higher than the yield strength of bainite [18]. In this sense, the strength and plasticity difference between ferrite and bainite is another key factor affecting the mechanical properties of F/B dual-phase pipeline steels. During the F/B pipeline steel TMCP processes, the ferrite transformation occurs during the cooling process, inthetime between the finishing of rolling and beginning of quick cooling, i.e., relaxation time. Then, the bainite is obtained during the subsequent rapid cooling [9]. Previous research shows that the finishing rolling temperature and beginning quick cooling temperature, i.e., relaxation time, are very important in determining the volume fractions of ferrite and bainite [12,13]. The C content in undercooled austenite increases as the ferrite transformation and amount increases; simultaneously, the subsequent rapid cooling rate affects the microstructure and mechanical properties of the bainite [19]. As a result, the bainite change in strength and plasticity causes the variation in the strength and plasticity of the F/B dual-phase pipeline steels. Therefore, the difference between ferrite and bainite is another critical factor affecting the mechanical properties of F/B dual-phase pipeline steels. Nevertheless, few studies exist on the effect of the difference between ferrite and bainite on the strength and deformability of the F/B dual-phase pipeline steels.

In this work, abstract representative volume elements (RVE) of FEM models were developed based on the mesostructure of the F/B dual-phase pipeline steel. Subsequently, the effects of ferrite and bainite on the F/B dual-phase pipeline steel tensile properties were investigated. Moreover, the strain and stress distributions of ferrite and bainite in the mesostructure of F/B steel were simulated, and the difference in tensile strength between the ferrite and bainite on the strain localization factor (SLF) is discussed. This work aims to reveal the mechanism by which the difference in tensile strength between ferrite and bainite affects the deformation compatibility in the mesostructure and to help optimize the microstructure of the F/B dual-phase pipeline steels to obtain strength and deformability balance.

## 2. FEM Model Generation and Analysis

Previous works have shown that the representative volume elements (RVE) based on mesostructure and abstract RVE models are effective methods for studying the mechanism of plastic deformation behavior of multi-phase steel [20,21]. According to our previous work [16,17], the F/B dual-phase X80 pipeline steel microstructure consists of elongated bainite distributed in a banded ferrite matrix with a bainite volume fraction of 41% (Figure 1a). The mesostructure-based abstract RVE model was built as shown in Figure 1b. In the abstract RVE model, 40% bainite volume fraction was selected because the optimal volume fraction of bainiteranges from 40% to 48% [16].

Since it is challengingto obtain the mechanical properties of individual phases in the microstructure with the same volume fraction of bainite, based on the previous results and considering the hardness range of bainite, the mechanical properties of the individual phases of ferrite and bainite were simulated and determined, as shown in Table 1. The five models correspond to different bainite strengths. In the models, the Young’s modulus (*E*) and Poisson’s ratio (*μ*) are assumed to be the same, while strain hardening index (*n*) and yield strength ($\sigma_{y,B}$) are emphasized. The yield strength ratio of bainite to ferrite ($\sigma_{y,B}/\sigma_{y,F}$) was determined to be from 2.11 to 3.30. In comparison, the actual F/B dual-phase X80 pipeline steel $\sigma_{y,B}/\sigma_{y,F}$ is about 2.3.

Finite element analysis shows the direct distribution of stress and strain in the mesostructure. In this work, the microscopic stress and strain in the singlephase were calculated by the method in reference [22]. After obtainingthe stress and strain in the single phase, the mesostructure stress and strain of abstract RVE were calculated by the method in reference [23].

## 3. Results

### 3.1. Mechanical Properties of Models with Different $\sigma_{y,B}/\sigma_{y,F}$

The stress-strain curves and mechanical properties of the models with a bainite to ferrite yield strength ratio of ($\sigma_{y,B}/\sigma_{y,F}$) obtained by the simulation are shown in Figure 2. The yield strength ($\sigma_{y}$) and tensile strength ($\sigma_{t}$) increase with the increase in $\sigma_{y,B}/\sigma_{y,F}$, where the yield strength ($\sigma_{y}$) increase was higher than the yield strength ($\sigma_{t}$). As the $\sigma_{y,B}/\sigma_{y,F}$ increases from 2.11 to 3.20, the tensile strength increases from 755 MPa to 859 MPa, an increase of 13%; meanwhile, the yield strength increases from 565 MPa to 668 MPa, for an 18% increase. As a result, the yield ratio (yield strength/tensile strength) increases from 0.748 to 0.778. Meanwhile, with the same $\sigma_{y,B}/\sigma_{y,F}$ increase, the uniform elongation (UE) decreases from 10.5% to 8.8%.

### 3.2. Effects of $\sigma_{y,B}/\sigma_{y,F}$ on Strain Hardening Behavior

The results, as stated above, show that the difference inmechanical properties between bainite and ferrite can notablyaffect the strength and plasticity of F/B dual-phase pipeline steel. Thisphenomenon may be attributed to the influence of $\sigma_{y,B}/\sigma_{y,F}$ on the deformation compatibility of the two phases during deformation. The plasticity of steelrelates to the strain hardening behavior. Therefore, the strain hardening behaviors of the models with different $\sigma_{y,B}/\sigma_{y,F}$ were analyzed by Hollomon and modified Crussard-Jaoul analyses.

The Hollomon analysis is one ofthe easiest ways to analyze the strain hardening ability of a material. The relationship of actual stress and actual strain curves of metal materials can be described as [24]:(1)$$\sigma =K\varepsilon^{n}$$
where σ is the true stress, ε is the actual strain, n is the strain hardening exponent, and K is the hardening coefficient. Taking the logarithm of both sides of Equation (1), it can be expressed as:(2)lnσ=nlnε+lnK

Equation (2) shows there is a linear relationship between *lnσ* and *lnε*, and the slope of the *lnσ* − *lnε* curve is the strain hardening exponent in the Hollomon analysis. The *lnσ* − *lnε* curves of the models with different $\sigma_{y,B}/\sigma_{y,F}$ are shown in Figure 3a.The strain hardening exponentgenerally refers to the plastic deformation stage; thus, the slope of the relationship curves of *lnσ* vs. *lnε* in the uniform plastic deformation stage is taken, and then the effect of $\sigma_{y,B}/\sigma_{y,F}$ on the strain hardening exponent of each model is obtained, as shown in Figure 3b. The strain hardening exponent of the models decreases with the increase in $\sigma_{y,B}/\sigma_{y,F}$, meaning that the uniform elongation decreases, as shown in Figure 2b.

The Hollomon analysis provides the relationship of strain hardening exponent and deformability with the $\sigma_{y,B}/\sigma_{y,F}$ for the F/B dual-phase pipeline steels; however, it cannot explain the micro-mechanism of effect $\sigma_{y,B}/\sigma_{y,F}$ on the strain hardening behaviorfor the F/B dual-phase pipeline steelmodels. Therefore, modified C-J analysis was used in this work, as it has been proved to be sensitive for characterizing the strain hardening capability [25]. The modified C-J analysis and the deduced form are shown in Equations (3) and (4) [25]:(3)$$\varepsilon_{p}=\varepsilon_{0}+C\sigma^{m}$$
(4)$$ln\left(\frac{d\sigma }{d\varepsilon_{p}}\right)=\left(1-m\right)ln\sigma -lnCm$$
where σ is the true stress, $\varepsilon_{p}$ is the actual plastic strain, $\varepsilon_{0}$ is the initial true strain, m is the inverse of strain hardening exponent, and C is the material constant. The relationship between lnσ with $ln\left(\frac{d\sigma }{d\varepsilon_{p}}\right)$ was calculated as shown in Figure 4a. The strain hardening behavior of models with different $\sigma_{y,B}/\sigma_{y,F}$ can be divided into three stages. Each stage is associated with the dislocation evolution and deformation for each phase [26]. Stage I is characterized by dislocations glide, entanglement, and concentration in ferrite near the F/B interface. With increasing load, the dislocations and slip systems in bainite are activated and propagated; as a result, uniform deformation occurs in ferrite and bainite, which is the characteristic of stage II. In stage III, the strain hardening ability of the steel decreases because some micropores begin to form, and the flow stress begins to decrease; thus, the flow stress is unaltered by dislocations [26,27].

Modified C-J analysis provides the slope of each stage by linear fitting, and the slope is 1-m, which is used to describe the strain hardening capability of the models. The higher value of 1/m represents a higher strain hardening capability [26,27]. The strain hardening capacity of each stage in the modified C-J analytical model is shown in Table 2. From Table 2, the 1/m value in stage II is higher than stagesI and III. Therefore, the deformation in stage II determined the deformability of the models. In addition, with the increase in $\sigma_{y,B}/\sigma_{y,F}$ of models, the value of 1/m representing the strain hardening capability decreases, consistent with the results from the Hollomon analysis (Figure 3b).

The $\sigma_{y,B}/\sigma_{y,F}$ effects on the transition strain from stage I to II, and from II to III of the models are shown in Figure 4b. The difference in theeffect of $\sigma_{y,B}/\sigma_{y,F}$ on the transition strain from stage I to II, and transition strain from II to III can be determined. With the $\sigma_{y,B}/\sigma_{y,F}$ increase, the transition strain from stage I to II increases, while the transition strain from stage II to III slightly decreases.

According to the deformation mechanism of dual-phase steel, the deformation always occurs in the soft matrix phase, and then the stress transfers into the secondary hard phase [26]. The efficiency of strain transition from stage I to II helps reduce the discrepancy of plastic deformation between the soft phase and the hard phase. The low transition strain means that the co-deformation can take place earlier. Therefore, the discrepancy of plastic deformation between the ferrite and bainite in the model increases when $\sigma_{y,B}/\sigma_{y,F}$ is higher (Figure 4b) because it needs more deformation in ferrite to generate more strain and stress concentration near the F/B interface bainite deformation. Conversely, the higher strain and stress concentration near the F/B interface increases the tendency of micro-crack initiation near the F/B interface, resulting in the strain transition from stage II to III decreasing (Figure 4b); consequently, the uniform deformation decreases. Therefore, the strain and stress concentration in the models should be considered to understand the effect of $\sigma_{y,B}/\sigma_{y,F}$ on the deformation mechanism of the F/B dual-phase X80 pipeline steel.

### 3.3. Effect of $\sigma_{y,B}/\sigma_{y,F}$ on Strain Localizationin the Models

Figure 5 shows the distribution of equivalent strain variation within the intermediate structure as applied load up to tensile strength, i.e., the applied strain reaches maximum uniform strain. The distribution of equivalent strain within the mesostructure is ununiform. The max shear strain bands along a 45° angle to the loading direction appear in the mesostructure; meanwhile, the strong strain localization occurs in the ferrite near the interface between ferrite and bainite. The $\sigma_{y,B}/\sigma_{y,F}$ does not affect the distribution characteristics of equivalent strains within the mesostructure because the models have some bainite volume fraction, whereas the $\sigma_{y,B}/\sigma_{y,F}$ affects the strain localization between ferrite and bainite. With the increase in $\sigma_{y,B}/\sigma_{y,F}$, the difference inaverage and localized equivalent plastic strain between the ferrite and bainite exhibits an increasing tendency. The statistical results of the probability distribution function (PDF) calculated from Figure 5 confirm this.

Figure 6 shows the PDF of equivalent plastic strain using the calculations shown in Figure 5. The $\sigma_{y,B}/\sigma_{y,F}$ has a minor effect on the strain distribution of ferrite. Nevertheless, the range of strain distribution in bainite diminishes with the increase in $\sigma_{y,B}/\sigma_{y,F}$. When $\sigma_{y,B}/\sigma_{y,F}$ is 2.1, the strain in bainite distributes in the range from 0 to 0.2, while the strain of over 40% bainite is concentrated at about 0.1. The strain in ferrite distributes in the range from 0 to 0.3, and the strain of about 30%ferrite is distributed at about 0.13. Whenthe $\sigma_{y,B}/\sigma_{y,F}$ is 3.1, the strain in bainite distributes in the range from 0 to 0.2, and the strain of over 65% bainite is about 0.05. Meanwhile, the strain in ferrite remains distributed in the range from 0 to 0.3, while the strain of about 30% ferrite is distributed at approximately 0.05. From the simulation and PDF of equivalent plastic strain results (Figure 6), it is assumed that the $\sigma_{y,B}/\sigma_{y,F}$ in models affect the plastic strain localization between the ferrite and bainite, affecting the mesostructure’s deformation compatibility. Thus, the results can be analyzed by the strain localization factor (SLF). The SLF can quantitatively reveal the strain localization and deformation compatibility between two phases in dual-phase steel. The SLF was defined as follows [28]:(5)$$\mathrm{SLF}=\overset{n}{\underset{i=1}{\sum }}\left(\varepsilon_{i,F}F_{i,F}-\varepsilon_{i,B}F_{i,B}\right)$$
where $\varepsilon_{i,F}$ and $\varepsilon_{i,B}$ are the equivalent plastic strain of *i*-th bin for ferrite and bainite, respectively, and the $F_{i,F}$ and $F_{i,B}$ are the area frequency of $\varepsilon_{i,F}$ and $\varepsilon_{i,B}$, respectively. The low value of SLF corresponds to the lower difference in plastic strain between different phases, while the high value of SLF corresponds to the increasing difference in plastic strain between different phases.

The $\sigma_{y,B}/\sigma_{y,F}$ effect on the SLF is shown in Figure 7. The SLF increases with the increase in $\sigma_{y,B}/\sigma_{y,F}$. The results indicate that the increasing $\sigma_{y,B}/\sigma_{y,F}$ will enhance the strain localization in mesostructure, as illustrated in Figure 8. Figure 8 shows the strain distribution along with the deformed band in Figure 5. The deformed band was considered the area with the most intense deformation. Moreover, the strength difference between bainite and ferrite in mesostructure will intensify the uneven strain distribution. The largest equivalent strain appears in ferrite in the front of the bainite corner, and the degree of strain localization in this location significantly increases with the increase in the $\sigma_{y,B}/\sigma_{y,F}$ in models, as shown in Figure 8b. The localized strain increases from 0.6 to 1.4 when the $\sigma_{y,B}/\sigma_{y,F}$ rises from 2.11 to 3.20. The higher strain localization facilitates the micro-crack initiation at this location and decreases the deformability.

## 4. Discussion

The simulated results for the F/B dual-phase pipeline steels show that the yield strength difference between bainite and ferrite greatly affects the tensile properties. The yield strength, tensile strength, and yield ratio increase with the $\sigma_{y,B}/\sigma_{y,F}$ increase in models, while the uniform elongation decreases (Figure 2). In addition, the strain hardening exponent as calculated by the Hollomon analysis also decreases (Figure 3). The effect of the difference between bainite and ferrite ($\sigma_{y,B}/\sigma_{y,F}$) on the tensile properties of F/B dual-phase pipeline steels is attributed to the deformation compatibility between the soft phase of ferrite and hardness phase of bainite.

For the F/B dual-phase steels, as the load is applied during the tensile deformation process, the deformation occurs in the soft phase of ferrite, i.e., the dislocations firstly glide under maximum shear stress along an angle of 45° to loading direction [29,30]; this is why the deformation bands appear in the models (Figure 5). Bainite will inhibit the dislocations glide, and the dislocations are arranged and entangled in ferrite in front of the grain boundary between bainite and ferrite. As a result, the strain and stress concentration accumulate in this region, and the deformation in ferrite is restrained [18]. With the increasing load, the accumulated strain and stress increase. As the accumulated stress is higher than the yield strength of bainite, the dislocations and slip systems in bainite are activated and propagated; as a result, uniform strain or deformation of the soft phase of ferrite and hard phase of bainite continues in either a coordinated or uncoordinated manner, and overall yielding in F/B dual-phase steels occurs [26,31]. In this case, the critical stress and strain to activate bainite deformation in models will increase with the increasing yield strength of bainite. Thus, it suggests that the yield strength difference between bainite and ferrite may affect the transition strain from single ferrite deformation to two-phase uniform deformation and the yield strength of overall yield in two phases. The suggestion agrees with the simulated results in this work. As the $\sigma_{y,B}/\sigma_{y,F}$ increases from 2.11 to 3.20, the transition strain from single ferrite deformation to two-phase uniform deformation increases from 2.0% to 4.2% (Figure 4b); consequently, the yield strength of the model for the F/B dual-phase steel increases from 565 MPa to 668 MPa (Figure 2).

Moreover, the increasing yield strength of bainite affects the critical transition strain from a single-phase ferrite deformation to two-phase uniform deformation in models, the deformation compatibility between ferrite and bainite, and the deformability of F/B dual-phase steels. As the deformation enters the uniform deformation stage, although the bainite also begins to deform, the incoordinate deformation between bainite and ferrite still exists in the model because the bainite has a low plasticity and strain hardening exponent. Greater plastic deformation of ferrite than bainite is required to support the deformation of bainite continuously. Therefore, with tensile deformation increase, the deformation difference between ferrite and bainite in the models increases (Figure 5b). Moreover, increasing the strength of bainite also promotes the deformation difference between ferrite and bainite in the models. As shown in Figure 5 and Figure 6, even though the applied strain almost reached maximum uniform deformation, the plastic deformation mainly remains concentrated in the ferrite. Meanwhile, the deformation difference between ferrite and bainite in the models increases with $\sigma_{y,B}/\sigma_{y,F}$ increase; as a result, the SLF increases (Figure 7). On the other hand, increasing the $\sigma_{y,B}/\sigma_{y,F}$ restricts the plastic deformation in ferrite between the two bainite grains, the PDF of equivalent plastic strain abates (Figure 6), and the severe strain localization occurs in ferrite in front of the bainite grain corner to coordinate the plastic deformation (Figure 8).

Many researchers believe that the tensile fracture mechanism of dual-phase steels is related to the initiation, growth, and connection of micro-voids formed during the deformation [29,30]. Liu et al. [18] found that the crystal slip in ferrite occurs relatively more easily during plastic deformation. At the same time, the stress concentration at the ferrite/bainite interface drives bainite rotation to accommodate the deformation of ferrite. Thus, voids are initiated at the F/B interface or in ferrite when the stress concentration satisfiesthe fracture strength criteria. Meanwhile, the high-stress concentration in bainite may lead to initiate voids and to bainite fracture. From this point of view, with the increase in bainite strength in the models, the bainite is difficult to rotate to accommodate the deformation of ferrite under the stress concentration. The high strain and stress concentration in ferrite near the ferrite/bainite interface is facilitated (Figure 8), and the initiation of micro-voids is accelerated. Hence, the uniform elongation of F/B dual-phase steel decreases with the increase in the $\sigma_{y,B}/\sigma_{y,F}$ (Figure 2).

In the results stated above, the difference in strength between ferrite and bainite in the models strongly affects the strength and plasticity of F/B dual-phase pipeline steels. With the bainite strength increase, the strength and yield ratio of F/B dual-phase pipeline steels significantly increase, while the strain hardening exponent and uniform elongation decrease. The effect of change in the strength difference between ferrite and bainite on the tensile properties for F/B dual-phase pipeline steels is attributed to the difference in deformation compatibility of two phases. Increasing the strength of bainite increases the critical stress of the dislocations glide in bainite and the transition strain from single-phase ferrite deformation to two phases of uniform deformation; thus, the yield strength is improved. On the other hand, with the increasing strength difference between ferrite and bainite, the ferrite deformation between the two elongated bainite grains is restricted, and the strain localization factor between ferrite and bainite increases, mainly in areas of stress and strain concentration, which may accelerate the micro-voids formation and decrease the uniform elongation. The results suggest that the volume fraction and morphology of bainite may be other factors to affect the deformation of ferrite and the strain localization in ferrite near the bainite [16,17]. Therefore, the combined influence of strength, volume fraction, and morphology of bainite on the strength, yield ratio, strain hardening exponent, and uniform elongation of F/B dual-phase pipeline steels should be researched further.

## 5. Conclusions

The yield strength difference between bainite and ferrite (namely, the ratio of bainite to ferrite ($\sigma_{y,B}/\sigma_{y,F}$)) in the mesostructure for the F/B DP steels on deformation compatibility has been studied by abstract RVE models. The main conclusions from this work are as follows:For mesostructure of the F/B DP steel with different $\sigma_{y,B}/\sigma_{y,F}$, the yield and tensile strength increase from 565 MPa and 755 MPa to 668 MPa and 859 MPa, respectively, when the $\sigma_{y,B}/\sigma_{y,F}$ increases from 2.11 to 3.20. Simultaneously, the yield ratio increases from 0.748 to 0.778; consequently, the uniform elongation decreases from 10.5% to 8.8%.Increasing the strength of bainite enhances the critical stress required for dislocation glide in bainite. As a result, the transition strain from single-phase ferrite deformation to simultaneous deformation of ferrite and bainite, i.e., the strain at which uniform deformation begins, increases the yield strength.Increasing the strength of bainite increases the strain localization factor (SLF), especially the local strain, and decreases the deformation compatibility of mesostructure, which decreases the strain hardening exponent and increases the tendency of micro-voids formation; therefore, the uniform elongation decreases.

## Figures and Tables

**Figure 1 materials-14-05352-f001:**
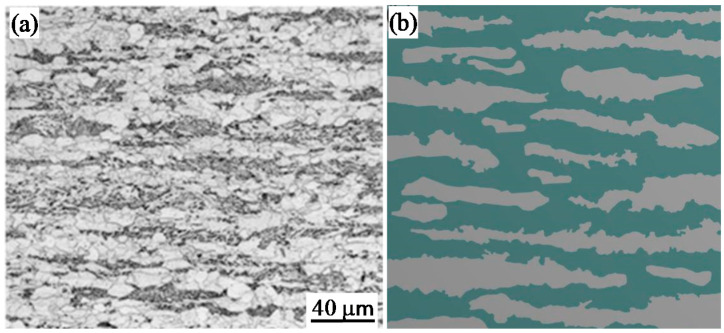
(**a**) Optical microstructure of dual-phase X80 pipeline steel and (**b**) abstract RVE model based on mesostructure.

**Figure 2 materials-14-05352-f002:**
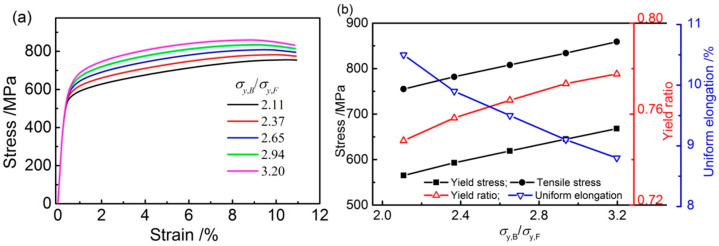
Simulated results of effect of $\sigma_{y,B}/\sigma_{y,F}$ on (**a**) stress-strain curves, and (**b**) mechanical properties of mesostructure.

**Figure 3 materials-14-05352-f003:**
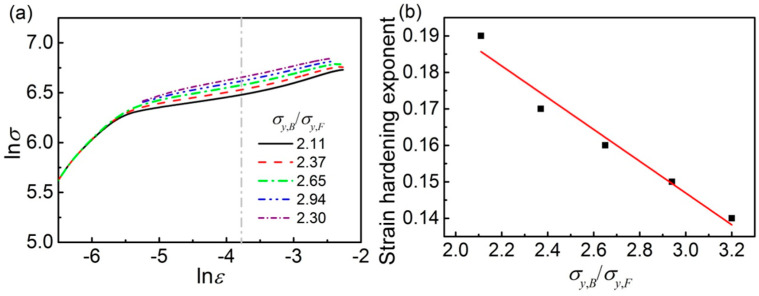
Influence of the $\sigma_{y,B}/\sigma_{y,F}$ in mesostructure on (**a**) relationship of *lnσ* vs. *lnε* analyzed by Hollomon analysis, and (**b**) effect of $\sigma_{y,B}/\sigma_{y,F}$.

**Figure 4 materials-14-05352-f004:**
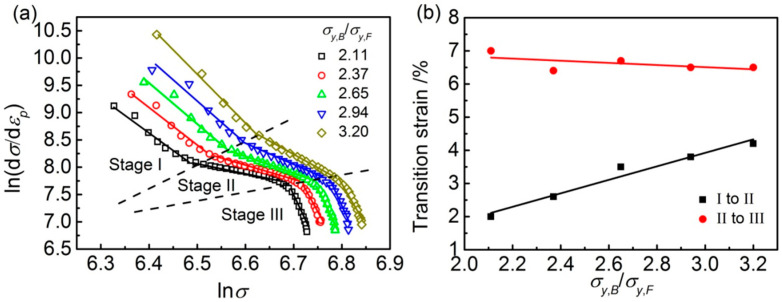
(**a**) Modified C-J analysis of models, and (**b**) transition strain vs. $\sigma_{y,B}/\sigma_{y,F}$ of the models.

**Figure 5 materials-14-05352-f005:**
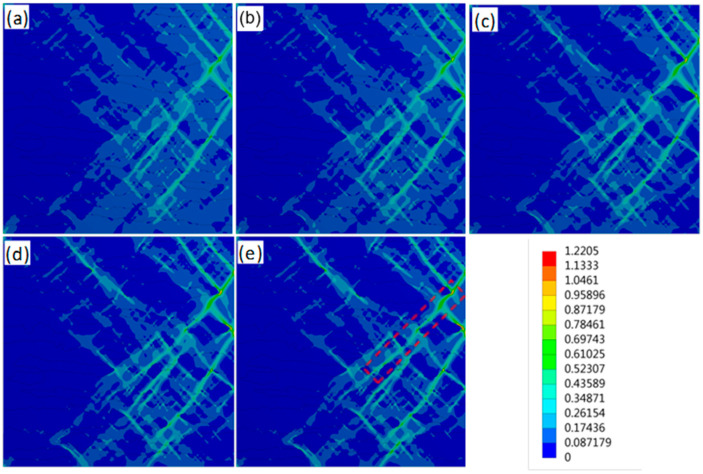
Distribution of equivalent strain within mesostructures as applied load up to tensile strength of different models with $\sigma_{y,B}/\sigma_{y,F}$ of (**a**) 2.11, (**b**) 2.37, (**c**) 2.65, (**d**) 2.94, and (**e**) 3.20.

**Figure 6 materials-14-05352-f006:**
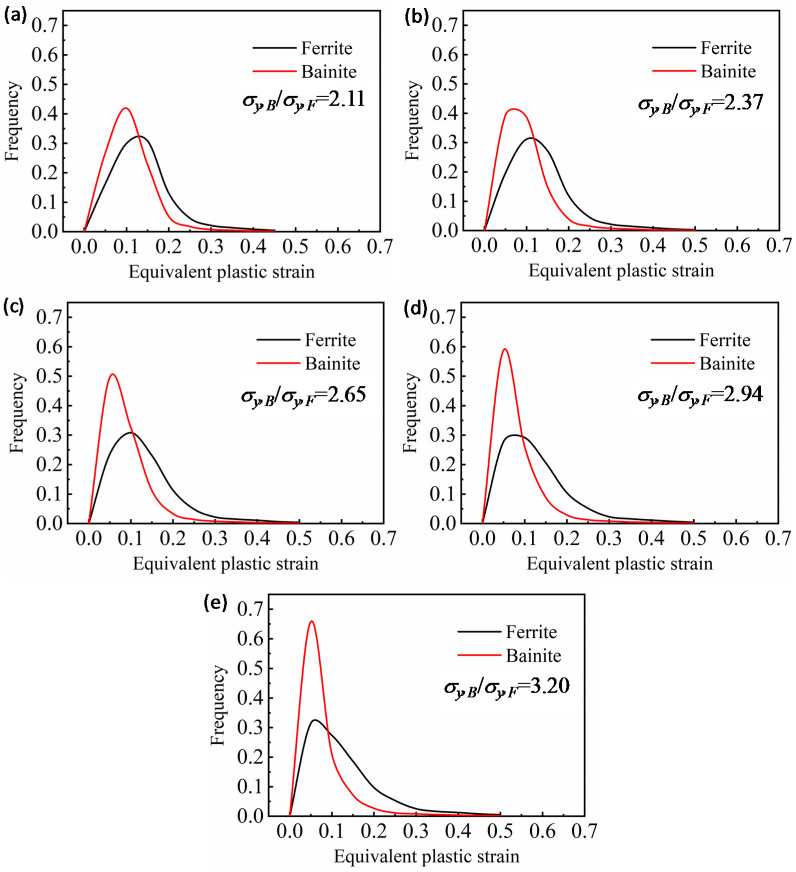
PDF curves of equivalent plastic strains of ferrite and bainite in the model with $\sigma_{y,B}/\sigma_{y,F}$ of (**a**) 2.11, (**b**) 2.37, (**c**) 2.65, (**d**) 2.94, and (**e**) 3.20.

**Figure 7 materials-14-05352-f007:**
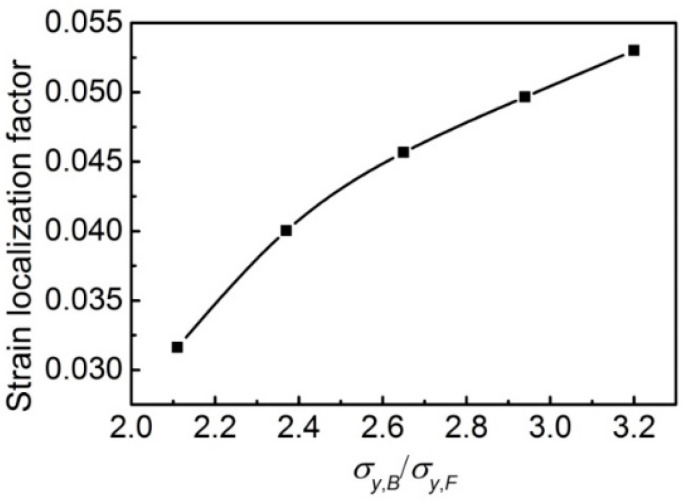
SLF vs. $\sigma_{y,B}/\sigma_{y,F}$ of the models.

**Figure 8 materials-14-05352-f008:**
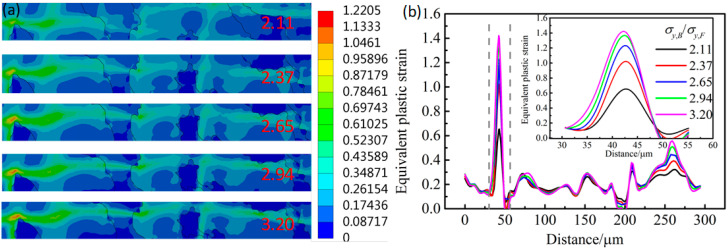
(**a**) Distribution of in-strain bands in Figure 5 and (**b**) line distribution of equivalent stain along the strain bands in models with different $\sigma_{y,B}/\sigma_{y,F}$.

**Table 1 materials-14-05352-t001:** Parameters of mechanical properties of bainite and yield strength ratio of bainite to ferrite.

Models	E (GPa)	μ	*n*	$\sigma_{y,B}\;(\mathrm{MPa})$	$\sigma_{y,B}/\sigma_{y,F}$
A	190	0.300	0.069	780	2.11
B	190	0.300	0.062	863	2.37
C	190	0.300	0.056	976	2.65
D	190	0.300	0.051	1087	2.94
E	190	0.300	0.047	1163	3.20

**Table 2 materials-14-05352-t002:** Strain hardening capability of models.

Model	$\sigma_{y,B}/\sigma_{y,F}$	Stage I		Stage II		Stage III	
		1−m	1m	1−m	1m	1−m	1m
A	2.11	−6.481	0.134	−2.158	0.317	−18.059	0.052
B	2.37	−7.807	0.114	−2.618	0.276	−13.498	0.069
C	2.65	−7.250	0.121	−3.281	0.234	−17.684	0.054
D	2.94	−7.124	0.123	−4.469	0.183	−18.067	0.052
E	3.20	−8.771	0.102	−4.896	0.170	−17.352	0.054

## Data Availability

The data can be requested from the corresponding authors.

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
