# Peer review of "Effect of Bainite to Ferrite Yield Strength Ratio on the Deformability of Mesostructures for Ferrite/Bainite Dual-Phase Steels"

_materials, 2021, doi:10.3390/ma14185352_

Round 1

Reviewer 1 Report

The paper reports modelling of plastic behaviour of dual (ferritic-bainitic) steel. It can be published if the mandatory points are improved and numerous minor remarks are considered.

Mandatory revision

  • Make clear in the abstract that the paper consists in modelling and no experimental data are given. E.G. in line 18: ..."first. IN THIS PURELY THEORETICAL PAPER, an abstract ...
  • Explain in more details the modelling procedure
  • Explain better what happens in the stage III

Minor revisions

  • Line 25 … „ferriteand“ … space is missing
  • Line 26-27 … „localised strain localisation.“ Strange, reformulate the phrase.
  • Line 60 … „onthe“ … space missing
  • Line 74 „ … that the TIME BETWEEN the finishing of rolling and beginning of quick cooling, i.e. relaxation time …“ Is it what you mean?
  • Line 82 Phrase not finished
  • Line 109 …“phases of ferrite AND BAINITE? were simulated“ …
  • Line 112 Table 2 (this is the first table in the paper, so it should be Table 1) is very unclear. What is „n“? What are the differences in lines A-E? How the yield stresses were calculated? Or the values were chosen? The ferrite is assumed to have the same yield stress (give the value) and properties of bainite vary? There are refs to papers [22] and [23], however at least brief description of the method should be given here as well as assumptions behind the calculations.
  • Line 144 … sigma is the actual TRUE? Stress, epsilon iis the actual TRUE? strain
  • Line 157 – … „thesigma“… space is missing
  • Line 170 space missing after „ln sigma“
  • Line 177 – I do not understand explanation of stage III
  • Line 179 space is missing after „strain“
  • Line 186 … representing, not „represented“
  • Line 208 Fig. 5, not 6
  • Line 208 ….“equivalent STRAIN“
  • 5 – The distribution of phases corresponds to Fig. 1b? Give this information in the description of Fig. If not, show the initial microstructure.
  • Line 209 Was the model loaded to 10% strain (line 209) or up to the ultimate tensile stress (or tensile strength) as specified in the Fug. 5 caption?
  • Line 212 The F-B interface (not grain boundary) are not visible in Fig. 5
  • Line 222 …“strain USING THE CALCULATIONS SHOWN IN Fig.5.
  • Lines 226, 227, 230, 235, 238, 285, 288, 344 space is missing
  • Line 257 … in Fig. 5, not 6
  • Line 239 Give practical explanation what SLF means: „The low value of SLF corresponds to the ……, while the high value of SLF ….“ Does it describe the difference of plastic strain between phases or nonuniformity of plastic strain in individual phases?
  • Line 261 …“difference between the YIELD STRESS of B and F“…
  • Line 271 „inhibit“ not inhibits
  • Line 275 what do you mean by „accumulated“? Local (or Concentration of) stress and strain close to the F-B interface increases?
  • Line 340 I prefer „The YIELD STRESS difference …“
  • Line 353 reformulate „the localised strain localisation“

Author Response

Comments and Suggestions for Authors

The paper reports modelling of plastic behaviour of dual (ferritic-bainitic) steel. It can be published if the mandatory points are improved and numerous minor remarks are considered.

Mandatory revision

  1. Make clear in the abstract that the paper consists in modelling and no experimental data are given. E.G. in line 18: ..."first. IN THIS PURELY THEORETICAL PAPER, an abstract ...

---Thank reviewer for their suggestions. We have revised in the manuscript.

Explain in more details the modelling procedure

--- Thank reviewer for their suggestions. We have revised in the manuscript.

Explain better what happens in the stage III.

--- Thank reviewer for their suggestions. We have revised in the manuscript.

Minor revisions

Line 25 … “ferriteand“ … space is missing

--- It has been revised.

Line 26-27 … “localised strain localisation.” Strange, reformulate the phrase.

--- It has been revised to “the local strain concentration”.

Line 60 … “on the” … space missing

--- It has been revised.

Line 74 “ … that the TIME BETWEEN the finishing of rolling and beginning of quick cooling, i.e. relaxation time …”Is it what you mean?

---Thank reviewer for their suggestions. We have revised in the manuscript.

Line 82 Phrase not finished

---Thank reviewer for their suggestions. We have revised in the manuscript.

Line 109 …“phases of ferrite AND BAINITE? were simulated” …

---Thank reviewer for their suggestions. We have revised in the manuscript.

Line 112 Table 2 (this is the first table in the paper, so it should be Table 1) is very unclear. What is „n“? What are the differences in lines A-E? How the yield stresses were calculated? Or the values were chosen? The ferrite is assumed to have the same yield stress (give the value) and properties of bainite vary? There are refs to papers [22] and [23], however at least brief description of the method should be given here as well as assumptions behind the calculations.

---Thank reviewer for their suggestions. In Table 1, the five models correspond to different bainite strengths. In the models, the Young's modulus (E) and Poisson's ratio (m) of bainite are assumed to be the same, while changes of strain hardening index (n) and yield strength ( ) of bainite are emphasized, thus, the yield strength ratio of bainite to ferrite ( ) was determined.

Line 144 … sigma is the actual TRUE? Stress, epsilon is the actual TRUE? Strain. ---Thank reviewer for their suggestions. We have revised in the manuscript.

Line 157 – … „thesigma“… space is missing

--- It has been revised.

Line 170 space missing after “ln sigma”

--- It has been revised.

Line 177 – I do not understand explanation of stage III.

---Thank reviewer for their suggestions. In stage III, the strain hardening ability of the steel decreased because some micropores begin to form, and the flow stress begin to decrease, thus the flow stress is unaltered by dislocations.

Line 179 space is missing after “strain”

---Thank reviewer. It has been revised.

Line 186 … representing, not “represented”

---Thank reviewer. It has been revised.

Line 208 Fig. 5, not 6

---Thank reviewer. It has been revised.

Line 208 ….”equivalent STRAIN”

---Thank reviewer. It has been revised.

Fig. 5 – The distribution of phases corresponds to Fig. 1b? Give this information in the description of Fig. If not, show the initial microstructure.

--- Thank reviewer for their suggestions. The distribution of phases corresponds to Fig. 1b. In Fig. 5, the interface between bainite and ferrite is marked as black line, which can be clearly shown in Fig. 8.

Line 209 Was the model loaded to 10% strain (line 209) or up to the ultimate tensile stress (or tensile strength) as specified in the Fug. 5 caption?

--- Thank reviewer for their suggestions. In Fig. 5, the distribution of equivalent strain within mesostructures as applied load up to tensile strength, i.e., the applied strain gets to maximum uniform strain.

Line 212 The F-B interface (not grain boundary) are not visible in Fig. 5

---Thank reviewer. It has been revised.

Line 222 …“strain USING THE CALCULATIONS SHOWN IN Fig.5.

---Thank reviewer. It has been revised.

Lines 226, 227, 230, 235, 238, 285, 288, 344 space is missing

---They have been revised.

Line 257 … in Fig. 5, not 6

---Thank reviewer. It has been revised.

Line 239 Give practical explanation what SLF means: „The low value of SLF corresponds to the ……, while the high value of SLF ….“ Does it describe the difference of plastic strain between phases or nonuniformity of plastic strain in individual phases?

---Thank reviewer for their suggestions. We have added the explanation what SLF means.

Line 261 …“difference between the YIELD STRESS of B and F“…

---Thank reviewer. It has been revised.

Line 271 “inhibit” not inhibits

--- Thank reviewer. It has been revised.

Line 275 what do you mean by “accumulated”? Local (or Concentration of) stress and strain close to the F-B interface increases?

---It has been revised.

Line 340 I prefer “The YIELD STRESS difference …”

---It has been revised.

Line 353 reformulate “the localised strain localization”

---It has been revised.

Reviewer 2 Report

The manuscript “Effect of bainite to ferrite yield strength ratio on the deformability of mesostructures for ferrite/bainite dual-phase steels” by Gui-ying Qiao et. al is devoted to the analysis of the deformation behavior of metallic materials consisting of two phases characterized by different yield strengths (?y). From this point, it shows the possibilities of modeling for a wide range of different two-phase functional materials.

In this study, the analysis was carried out for two-phase pipeline steels of X80 strength class, which are intended for the manufacture of pipelines operating under extreme deformation conditions. The latter provides the subject of the manuscript with the up to date practical demand in energetics.

Notes:

  1. The anisotropy of mechanical properties of low carbon pipeline steels have been investigated since 1980s, both after TMCP and subsequent treatments. These studies including recent ones have shown that the anisotropy of mechanical properties originates due to development of strong crystallographic texture in all phases (austenite, martensite, ferrite, bainite). Thus, if the modeling is carried out for pipeline steels this fact should be taken into account. Therefore, the authors should have commented on the modeling capabilities, which enable to take the anisotropy of mechanical properties into account.
  2. L109 - Table 1 is not mentioned in the text. Check if Table 2 is a correct reference here. It is also necessary to give a more complete commentary on the models designations and the principle of the parameters selection in the table.
  3. L126 (The following sentence apparently contains an actual error) - “As a result, the yield factor (yield stress / tensile strength) decreases from 0.748 to 0.778”, though it factually increases in the Figure (red line) as well.
  4. The gaps between the words are missing. Examples: L25 - a space in the "ferrite"; L60 - space required in "onthe"; L190 - space required for "deformation"; L235 - Space Required in "TheSLF can quantify the localization of the deformity and".

Author Response

Comments and Suggestions for Authors

The manuscript “Effect of bainite to ferrite yield strength ratio on the deformability of mesostructures for ferrite/bainite dual-phase steels” by Gui-ying Qiao et. al is devoted to the analysis of the deformation behavior of metallic materials consisting of two phases characterized by different yield strengths (?y). From this point, it shows the possibilities of modeling for a wide range of different two-phase functional materials.

In this study, the analysis was carried out for two-phase pipeline steels of X80 strength class, which are intended for the manufacture of pipelines operating under extreme deformation conditions. The latter provides the subject of the manuscript with the up to date practical demand in energetics.

Notes:

The anisotropy of mechanical properties of low carbon pipeline steels have been investigated since 1980s, both after TMCP and subsequent treatments. These studies including recent ones have shown that the anisotropy of mechanical properties originates due to development of strong crystallographic texture in all phases (austenite, martensite, ferrite, bainite). Thus, if the modeling is carried out for pipeline steels this fact should be taken into account. Therefore, the authors should have commented on the modeling capabilities, which enable to take the anisotropy of mechanical properties into account.

---Thank reviewer’s good suggestions. Indeed, the anisotropy of mechanical properties is an important parameter for low carbon pipeline steels. However, in this paper, only 2D model were built, thus, the anisotropy was not considered. We will take the anisotropy of mechanical properties into account by 2D model in future work.

L109 - Table 1 is not mentioned in the text. Check if Table 2 is a correct reference here. It is also necessary to give a more complete commentary on the models designations and the principle of the parameters selection in the table.

---It has been revised.

L126 (The following sentence apparently contains an actual error) - “As a result, the yield factor (yield stress / tensile strength) decreases from 0.748 to 0.778”, though it factually increases in the Figure (red line) as well.

---It has been revised.

The gaps between the words are missing. Examples: L25 - a space in the "ferrite"; L60 - space required in "onthe"; L190 - space required for "deformation"; L235 - Space Required in "TheSLF can quantify the localization of the deformity and".

---It has been revised.

Reviewer 3 Report

The authors present an extensive and comprehensive study of the effect of the bainite / ferrite yield strength ratio on the deformability of mesostructures for ferrite / bainite dual-phase steels. The study will make an important contribution to our still scarce
knowledge of the mechanical behavior of F/B double phase and, in particular, of X80 pipeline steels. The object of study is well within the scope of Materials.
The introduction is largely well written, and the structure as well.
Overall, the methodology seems well thought out and solid, supported by a list of suitable references. The results section is also well presented. Some statistical analysis was performed, which
currently praises this article considerably. Furthermore, the discussion is also well supported by the results obtained. Apart from some writing errors throughout the text, requiring a general revision, I have nothing special to point to this article, recommending its publication after minor revisions.

Author Response

Comments and Suggestions for Authors

The authors present an extensive and comprehensive study of the effect of the bainite / ferrite yield strength ratio on the deformability of mesostructures for ferrite / bainite dual-phase steels. The study will make an important contribution to our still scarce knowledge of the mechanical behavior of F/B double phase and, in particular, of X80 pipeline steels. The object of study is well within the scope of Materials.

The introduction is largely well written, and the structure as well.

Overall, the methodology seems well thought out and solid, supported by a list of suitable references. The results section is also well presented. Some statistical analysis was performed, which currently praises this article considerably. Furthermore, the discussion is also well supported by the results obtained. Apart from some writing errors throughout the text, requiring a general revision, I have nothing special to point to this article, recommending its publication after minor revisions.

---Thank reviewer’s good suggestions. Some errors have been revised in manuscripts.

Reviewer 4 Report

The authors presented the influence of the bainite yield strength on the dual-phase pipeline steel. It is an interesting issue, but particular things should be explained. Authors should more precisely describe FEM analyse. Why the 10% strain was chosen? What mean dashed red lines on Fig. 5?

Particular comments are the following:

Line 100 It should be a reference to Fig. 1a).

Table 1There aren’t meanings of the symbols.

Line 208 Wrong reference, it is Fig. 6, but it should be Fig. 5.

Author Response

Comments and Suggestions for Authors

The authors presented the influence of the bainite yield strength on the dual-phase pipeline steel. It is an interesting issue, but particular things should be explained. Authors should more precisely describe FEM analyse. Why the 10% strain was chosen? What mean dashed red lines on Fig. 5?

---Thank reviewer’s good suggestions. We are very sorry, in FEM analysis, the applied strain was chosen when applied load was to tensile strength, i.e., applied strain was maximum uniform deformation. We have revised in the manuscript.

In addition, the dashed red lines on Fig. 5 is for clearer analysis of strain concentration, the results is illustrated in Fig. 6.

Particular comments are the following:

Line 100 It should be a reference to Fig. 1a).

---It has been revised.

Table 1 There aren’t meanings of the symbols.

---The meanings of the symbols have been shown in manuscript.

Line 208 Wrong reference, it is Fig. 6, but it should be Fig. 5.

---It has been revised.

Round 2

Reviewer 1 Report

The suggested revisions were considered and accepted, so the paper can be published according to my opinion.

Reviewer 4 Report

The manuscript could be published in present form.